# Non-Parametric Neighborhood Test-Time Generalization: Application to Medical Image Classification

Anonymous

Anonymous Institute
*******@***

**Abstract.** Reliable and stable performance is crucial for the application of computer-aided medical image systems in clinical settings. However, approaches based on deep learning often fail to generalize well under distribution shifts. In medical imaging, such distribution shifts can, for example, be introduced by changes in scanner types or imaging protocols. To counter this, test-time generalization aims to optimize a model that has been trained on single or multiple source domains to an unseen target domain. Common test-time adaptation methods fine-tune model weights utilizing losses with gradient-based optimization, a time-consuming and computationally demanding procedure. In contrast, our approach adopts a non-parametric method that is entirely feedforward and utilizes information from target samples to extract neighborhood information. By doing so, we avoid any fine-tuning or optimization procedures, which enables our method to be more efficient and achieve stable adaptation. We demonstrate the effectiveness of our approach by benchmarking it against different state-of-the-art methods with three backbones on two publicly available datasets, consisting of fetal ultrasound and retinal images. Our code is publicly available at: https://anonymous.4open.science/r/snviemroyxqz

**Keywords:** domain adaptation, generalization, unsupervised learning, parameter-free optimization

## 1 Introduction

Computer-aided medical imaging systems have achieved significant progress in recent years, with a substantial part of this progress made possible by the advancements of deep learning models [18,7,19,5]. However, a major limitation for their adoption in clinical environments is given by their restricted generalization capacity across unseen data distributions [29]. The reason for this is distribution shifts that can, for example, be caused by variations in scanner types or imaging protocols [20]. To address this, *test-time adaptation and generalization* arose as methods to optimize a trained source model on new incoming target data. Unlike *domain adaptation and generalization* techniques, test-time generalization can consecutively optimize the model on unlabelled data during the test phase without the requirement to access source data, fostering privacy-preserving adaptation to

the target domain. Additionally, it allows the model to be optimized continuously without interrupting the inference process, proving especially beneficial in time-sensitive applications where maintaining a flow of real-time decision-making is imperative. Furthermore, the capability of test-time generalization to process data in batches is reflective of real-world scenarios where medical data is also available serially. This aspect enhances its applicability in dynamic clinical workflows.

As shown in Figure 1(a), test-time generalization [24,21,15,14,17,8] methods focus on fine-tuning of the source model based on source model predictions, surrogate models or task predictions. This optimization often involves computation of gradients with norm-based losses followed by finetuning of batch norm layers [23], the whole model [14] or a linear classification layer [23]. A more recent approach [10] utilizes parameterized ensembles to optimize the last layers of the source model. Even though it is possible to only fine-tune the batch norm parameters [23], gradient-based fine-tuning of model weights, in general, is resource and time-intensive. This increases computational costs and leads to slower adaptation processes, making such methods less practical for real-time applications such as dynamic contrast-enhanced imaging, real-time tumor classification, and rapid stroke identification. Gradient-based finetuning [23,17,10,14] often relies on maximum a posteriori estimation to obtain the target model, which can lead to overfitting when adapting to target data that involves multiple distribution shifts. This reduces the model's generalizability, making it less robust to diverse shifts. Moreover, the potential to converge to local minima and a susceptibility to hyperparameter selections limits these methods efficacy.

By the nature of their feedforward design, non-parametric methods bypass overfitting and negate the need for loss-based gradient finetuning, therefore offering clear advantages, Fig. 1(b). T3A [9] computes class representations or prototypes based on the source model's weights and adjusts the classifier utilizing an entropy threshold. However, by relying only on the entropy of samples, information from the target domain is not fully utilized in the method. A more nuanced approach would be given by application of the source model to directly identify target samples with analogous characteristics to the source features. Such a method has the power to increase the utilization of target information and to align closely with the intrinsic data distribution.

Building on these insights, our work introduces a novel, non-parametric method coined Test-time Non-parametric Neighbors (TNN). We leverage neighborhood information without the need for finetuning. In summary, our contributions are:

- We propose utilizing target neighborhood information with dynamic voting to adjust source-trained classifiers in a non-parametric manner for test-time generalization.
- Our proposed method (TNN) is simple and does not modify the source training process. Yet, it is effective across datasets and requires minimal computation at test time due to its feedforward nature.
- We adopt several state-of-the-art test-time generalization techniques for medical imaging and perform exhaustive comparisons to our approach.

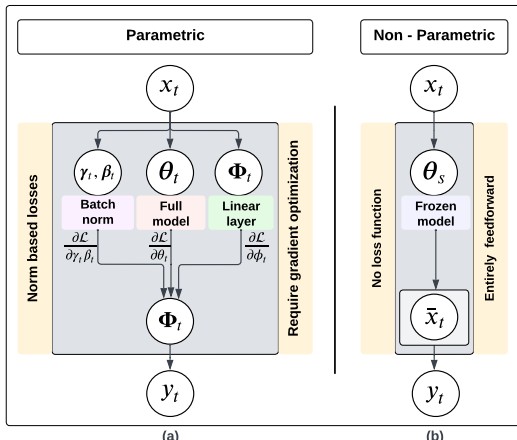

Fig. 1: **Data and model interaction scheme.** (a) Common test-time generalization techniques utilize norm-based losses for gradient-based finetuning of batch norm layers ($\beta_t$ and $\gamma_t$), full model ($\theta_t$), or linear layers ($\Phi_t$) to obtain target predictions $\mathbf{y}_t$. These methods feature memory and compute constraints and require precise hyperparameter selection with several rounds of backpropagation. (b) Non-parametric approaches obtain $\mathbf{y}_t$ via techniques that operate on frozen source model predictions $\bar{x}_t$ in a feed-forward manner. This neglects the need for additional computational resources and simplifies the generalization process.

Through comprehensive experiments and ablation studies, we demonstrate the efficacy and potential of TNN in medical imaging contexts, an area where such non-parametric approaches have been underexplored.

## 2 Background

Test-time domain generalization aims to generalize a model $\boldsymbol{\theta}_s$ trained on the source domains $\mathcal{S}$ to a unseen target domain $\mathcal{T}$, with $\mathcal{S}$ usually consisting of several source domains $\{D_s\}_{s=1}^{S}$. $\mathcal{T}$ may also consist of several target domains $\{D_t\}_{t=1}^{T}$. Here, $(\mathbf{x}_s, \mathbf{y}_s)$ and $(\mathbf{x}_t, \mathbf{y}_t)$ denote the image and corresponding label pairs on the source $\{D_s\}_{s=1}^{S}$ and target domain $\{D_t\}_{t=1}^{1}$, respectively. The objective of test-time domain generalization is to maximize the log-likelihood of the source model on the target data $p(D_t|\boldsymbol{\theta}_s)$, i.e., $p(\mathbf{y}_t|\mathbf{x}_t, \boldsymbol{\theta}_s)$.

**Formulation of parametric methods.** Due to distribution shifts between source and target domains, the source-trained model $\boldsymbol{\theta}_s$ is highly likely to fail on unseen target domains $D_t$, causing unreliable predictions with high confidence [1,27]. To prevent this, the source model must be generalized to the target domain at test time by transforming $\boldsymbol{\theta}_s$ to $\boldsymbol{\theta}_t$. Most common parametric methods

4 Anonymous

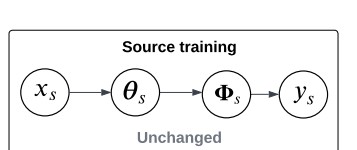 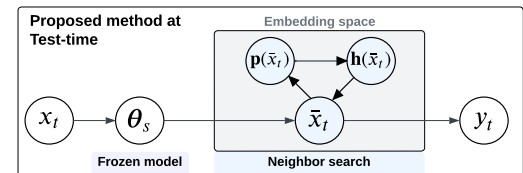

Fig. 2: **TNN at test-time.** We do not change the source training setup. In the embedding space, TNN performs neighborhood search $\mathbf{h}(\bar{x}_t)$ in a non-parametric manner after computing the prototypes $\mathbf{p}(\bar{x}_t)$ from the frozen source model. Followed by obtaining the classification label $\mathbf{y}_t$.

employ fine-tuning based on norm-based losses [10,14,23]. The log-likelihood of the target data is given by:

$$
\begin{aligned}
p(\mathbf{y}_t|\mathbf{x}_t,\boldsymbol{\theta}_s) &= \int p(\mathbf{y}_t|\mathbf{x}_t,\boldsymbol{\theta}_t)p(\boldsymbol{\theta}_t|\mathbf{x}_t,\boldsymbol{\theta}_s)d\boldsymbol{\theta}_t \\
&\approx p(\mathbf{y}_t|\mathbf{x}_t,\boldsymbol{\theta}_t^*),
\end{aligned}
\tag{1}
$$

with the integration of the distribution $p(\boldsymbol{\theta}_t)$ usually approximated by the maximum a posteriori (MAP) estimation. The final generalized MAP model $\boldsymbol{\theta}_t^*$ is obtained by fine-tuning of the parameters with one or multiple rounds of backpropagation using a norm-based unsupervised loss function, like entropy minimization [23], pseudo labeling [14] or task-specific losses [15,21]. However, fine-tuning the model parameters through gradient optimization makes parametric methods time-consuming and computationally expensive while also being sensitive to hyperparameter settings.

**Non-parametric methods.** To counter the above limitations, recent non-parametric methods such as [9] obtain class representations as prototypes, utilizing the weights of the source-trained linear classifier, i.e., without the need for MAP approximation or gradient-based optimization. Next, they obtain pseudo labels for the incoming target data based on the distance to those prototypes by applying entropy thresholds. After each incoming batch of target data, significant samples are selected, employing a threshold, and used to update the prototypes via simple adjustments to the classifier.

## 3 Method

**Source training.** Recent studies have shown that utilization of empirical risk minimization (ERM) [22,6] during source training enables models to generalize well under distribution shifts. Other methods, such as [26,28,1,25], included additional objectives to be minimized during source training. However, the requirement to interfere in the training procedure limits the applicability of such approaches. Therefore, we aim to develop a method that does not modify the

source training procedure, making it applicable to any pretrained model without any additional requirements. Specifically, as in [22,6], on multiple source domains $\{D_s\}_{s=1}^S$, given a source model $\boldsymbol{\theta}_s$, such as ResNet-18, and a loss function $\mathcal{L}$, such as cross-entropy, the total risk is minimized via $\mathbb{E}_{(x_s,y_s)\sim D_s}[\mathcal{L}(\theta_s(x_s),y_s)]$.

**Test-time generalization via nearest neighbors.** Our approach can be summarized by: at test-time, in a non-parametric way, we initially compute the source prototypes following [9]. Next, given a batch of target data, we obtain the nearest neighbors for classification and adjust the classifier weights as described below.

**Existence of nearest neighbors.** We propose that for a sufficiently trained model able to separate classes reasonably well in the source domain, cases that are similar in the higher dimensional image domain will lie close to each other in the source learned lower dimensional embedding space. This is ensured by the Johnson-Lindenstrauss (JL) [11] lemma as stated below:

Given a set of points $\{x_i : i = 1, ..., M\}$ in $\mathbb{R}^m$, the JL lemma states that if $n \geq c\epsilon^2 \log M$, with $0 < \epsilon < 1$, then there exists a linear map $A : \mathbb{R}^m \to \mathbb{R}^n$ such that for all $i \neq j$:

$$1 - \epsilon \leq \frac{\|A(x_i) - A(x_j)\|}{\|x_i - x_j\|} \leq 1 + \epsilon.$$

At test-time, utilizing the above lemma, our approach first computes the source model prototypes for each class in lower-dimensional space. To do so, as in [9], we initialize the class-specific prototypes by aggregating the weights of the source-trained linear classifier layer. When receiving new test-time data $\mathbf{x}_t$, we project it into lower-dimensional space using the source-trained model that preserves distances in the embedding space, as ensured by the JL lemma. In this embedding space, we assign $\mathbf{x}_t$ the label of its nearest class prototypes based on a distance measure (see below). Finally, we update the class-specific prototypes with the new sample to reflect new target characteristics. This allows us to classify new samples without the need for extensive computation or any optimization schemes to find the nearest points to the prototypes.

**Selecting neighbors with dynamic voting.** Since not all of the neighbors provide accurate information about the target data, i.e., some of them are noisy [4], we calculate the distance between the initial prototypes and new classifier weights obtained from incoming neighbor samples at test time for the selection of valid neighbors. We utilized the cosine distance here, while in principle every other distance metric can be use. Next, we use dynamic voting to obtain the most useful neighbors, i.e. we aggregate each neighbor's prediction and calculate the mean of the obtained new weights to determine the final weights of the classifier. When a new batch of samples arrives, the pseudo labels are predicted based on these classifier weights. This process is repeated iteratively for each new incoming batch of data.

## 4    Experiments

### 4.1    Datasets and implementation details

**Datasets.** We validate the effectiveness of TNN on two publicly available datasets. Messidor [3] depicts a retinal image database collected from three independent medical centers, containing 1200 images of diabetic retinopathy, each labeled with a severity score ranging from 0 to 3. Therefore, it consists of 3 domains, 4 classes, and 1200 images in total. Fetal-8 [2] is a maternal-fetal ultrasound dataset that consists of eight classes representing different anatomical planes collected from imaging scanners of two different vendors. Hence, it consists of 2 domains, 8 classes and 12,058 images in total. We consider the problem as a test-time generalization setting, not domain adaptation. Therefore, we train our model on multiple source domains of the Messidor dataset instead of only one. Furthermore, we follow the *leave-one out* evaluation standard from [9,13] and obtain the best source model following the training-domain validation split of [9]. For all the experiments, performance is reported on the target dataset with test-time adaptation [1,9,23] by utilizing accuracy as a metric.

**Implementational details.** On both datasets, Messidor and Fetal-8, we evaluate the performance of our approach based on three backbones, DenseNet-121, ResNet-18 and ResNet-50. Further in-depth ablation experiments are then performed with the ResNet-18 model. As in common test-time methods [9,6], the backbones are pretrained on Imagenet. Baseline and state-of-the-art methods are also implemented for the two datasets and for all of the three backbones, utilizing the Domainbed library [6] with all hyperparameters set to default. The training-validation split strategy of [9,6] is used for selection of the best source model. At test-time, we perform just forward passes to perform the classification on the online target data. We evaluate the performance of methods following standard test-time generalization evaluation [9]. We utilize a small batch size of 32 for test-time generalization, reflecting real-world practical scenarios.

### 4.2    Comparisons

We evaluate the performance of our model in reference to different state-of-the-art (SOTA) approaches and a source training strategy employing ERM minimization without adaption to the target domain as the baseline.

**State-of-the-art comparisons.** We compare our approach to existing parametric and non-parametric state-of-the-art methods by re-implementing them on the two datasets for all three backbones. Parametric methods, as shown in Figure 1, refer to techniques that finetune weights of the source-trained model, utilizing gradient optimization. Non-parametric methods refer to techniques that perform feedforward computation at test-time without any kind of finetuning, optimization, or usage of any external memory bank or an additional model.

Table 1 shows the performance comparisons. TNN achieves the best results on both datasets. Parametric methods achieve comparatively lower performance than the ERM baseline in many cases. One reason for this can be given by the fact that the medical imaging datasets at hand only contain a fraction of the number of samples the parametric methods were designed for. Furthermore, differences between images of distinct classes in the medical domain are way more subtle than in the computer vision domain, with different classes in the retinopathy images of Messidor even depicting a severity score that changes only gradually. Therefore, it is likely that non-parametric methods achieve better performance due to the fact that they do not fine-tune the complete source model weights but rather act upon the source-trained embedding space that should be able to separate classes reasonably well. T3A [9] utilizes the entropy of the samples as a threshold to classify new cases. In contrast, we use detailed neighborhood information for classification. For the Fetal-8 dataset, utilizing the ResNet-18 backbone, the performance of all the generalization methods decreases with reference to the ERM baseline. Reason for this is most likely overfitting due to the small size of the model, but also the small dataset size at test-time. For all the remaining settings, TNN performs better than the other approaches.

Table 1: **State-of-the-art comparisons** using DenseNet-121, ResNet-18, and ResNet-50 as backbones. We re-implement the methods on the datasets and report the mean accuracy across all the domains. The best results are in **bold.** Our results are averaged over five runs. Our method performs the best consistently with the highest performance improvement.

| | Messidor | | | Fetal-8 | | |
|---|---|---|---|---|---|---|
| **Methods** | DenseNet-121 | ResNet-18 | ResNet-50 | DenseNet-121 | ResNet-18 | ResNet-50 |
| ERM | 52.2 | 53.1 | 52.1 | 81.6 | 84.3 | 83.6 |
| *Parametric* | | | | | | |
| Tent [23] | 44.7 | 43.9 | 40.0 | 73.0 | 70.3 | 80.1 |
| SHOT [14] | 47.4 | 51.6 | 51.9 | 69.8 | 70.1 | 72.4 |
| ShotIM [14] | 47.5 | 51.8 | 46.0 | 68.9 | 69.9 | 72.0 |
| *Non-Parametric* | | | | | | |
| T3A [9] | 51.6 | 51.8 | 51.6 | 82.4 | 71.8 | 85.6 |
| *TNN (Ours)* | **53.1** $\pm$0.2 | **53.8** $\pm$0.2 | **53.9** $\pm$0.2 | **83.3** $\pm$0.2 | 72.7 $\pm$0.2 | **86.1** $\pm$0.2 |

### 4.3 Additional Experiments

**Addressing uncertain scenarios.** Ensuring alignment between model output probabilities and the actual likelihood of events is crucial in uncertain scenarios [12]. To quantify this alignment, Table 2 presents the expected calibration error [16] (lower values indicate better calibration) for our approach compared to the Tent model [23], utilizing a ResNet-18 backbone. Consistently, TNN demonstrates considerably better calibration scores across all domains on the Messidor

dataset. Therefore, leveraging its entirely feedforward nature for classifier adjustments, TNN achieves a better-calibrated model.

**Computational cost.** In Table 3, we compare the number of parameters required to be trained at test-time and the number of floating point operations per second (FLOPS) consumed by the GPU for Tent [23], T3A [9] and TNN. All of the methods, ours included, feature the same memory requirements of 1.4GB for the ResNet-18 model. However, as Tent optimizes the batch normalization layers of the target model at test time and, therefore, it requires more parameters to be trained than our approach. TNN and T3A are both non-parametric. Thus, they only perform a very limited amount of computational operations but do not need any additional computations to calculate the gradients on the GPU. Therefore, the measurement of TFlops is negligible in this case, considering the vast amount of computations required for weight optimization of parametric approaches. This is especially useful in limited resource settings.

Table 2: **Addressing uncertain scenarios.** ECE error on the three domains (0-2) of the Messidor dataset. The proposed method consistently reduces the ECE error across all the domains.

|  | ECE Error ↓ | | | |
|---|---|---|---|---|
|  | **0** | **1** | **2** | **Mean ↓** |
| Tent | 0.101 | 0.336 | 0.130 | 0.189 |
| **TNN (Ours)** | **0.001** | **0.005** | **0.003** | **0.003** |

Table 3: **Computational cost.** The number of new parameters to be trained at test-time alongside the TeraFlops consumed on the GPU. TNN and T3A both consume fewer resources and are thus useful for practical scenarios.

|  | **Parameters ↓** | **Model TFlops ↓** |
|---|---|---|
| Tent | 600000 | 212992 |
| T3A | 0 | - |
| **TNN (Ours)** | 0 | - |

## 5    Conclusion

We propose the usage of a non-parametric-based neighborhood classification method for medical imaging tasks that involve distribution shifts as a novel test-time generalization method. By utilizing target information and neighbors in the embedding space we sequentially adjust the weights of the classifier, providing an efficient yet powerful generalization technique. As with common methods, our method requires a shared label space between domains for adaptation. We aim to address this limitation in future work by enabling the generation of classifiers for new categories at test-time, alongside existing classifier adjustments. Furthermore, we also provide additional experiments to demonstrate the utility of the method in uncertain scenarios and settings that require limited computational resources.

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
