# OpenReview forum: "Non-Parametric Neighborhood Test-Time Generalization: Application to Medical Image Classification"
_MICCAI.org/2024/Workshop/MSB — MICCAI Student Board EMERGE Workshop 2024 Oral_

### Official Review · Reviewer_rJeC · 2024-07-05

**Recommendation:** 5
**Confidence:** 4

**Clarity:**

The paper is clear and well-written, with minor areas for improvement in clarity

**Feedback:**

- Abstract doesn't explicitly state WHAT method is applied and what results are to be expected. But these results can be pointed out with confidence, i.e. beating SOTA in terms of accuracy, runtime and ECE.
- The introduction is also rather vague on the idea behind TNN, and rather emphasizes the shortcomings of prior work. "We leverage neighborhood information without the need for finetuning." --> neighborhood of _what_?
- I'm not sure if Fig.1 is really needed after the thorough explanations in methods. I would rather like to see a visualization of the idea of taking the cosine distance between source prototypes and target samples.
- I had to consult [9] to get a better idea of source prototypes. Since the method relies heavily on these prototypes, it would be beneficial to include a brief description of the method to acquire such prototypes in the main manuscript.

**Justification:**

Interesting paper that presents a simple but effective method for domain adaptation. Major claims are based on a thorough evaluation. Flaws can be considered minor.

**Reproducibility:**

Sufficient amount of details available for reproducing the main results, and open access is provided (or promised upon acceptance) to source code and/or data

**Strengths:**

- The proposed approach is properly evaluated. Two medical datasets, three architectures each, five baseline methods (mix of parametric and non-parametric), and standard deviations for each result of the proposed method. All baselines are re-implemented.
- TNN outperforms the baselines in all cases, especially the other non-parametric approach.
- The paper is detailed, well-written and in most parts well-organized. The method is quite simple, but intuitive.
- Source code will be published, which is great.
- Clearly, a practical application of this approach is compelling.
- Authors did a good job describing and contextualizing related work.

**Summary:**

The authors propose TNN (Test-time Non-parametric Neighbors), a nearest-neighbor based parameter-free technique for the adaptation of neural network classifiers that doesn't require adaptation through training. During test time, source prototypes are generated for each class. Afterwards, the class is determined as argmax of the cosine distance between samples in embedding space. This method beats SOTA in terms of runtime and accuracy.

**Weaknesses:**

- Table 3 is redundant and confuses more than it adds to the paper. I think the point was already made clear in the very beginning of the manuscript that the proposed method doesn't require any training. Further, the comparison is slightly unfair, as only _Model_ TFLOPS are compared. Still, TNN will use some FLOPS to work (likely not in the range of TFLOPS), but these are not reported.
- How is the calibration score estimated? Why is it not reported for T3A? Why only for ResNet-18 and not all baseline models? It looks like the evaluation of ECE is not as complete as it could be.

---

> ### Author Response · Authors · 2024-07-12
> **Rebuttal by Authors**
>
> Thank you for recognizing the strengths of our proposed method and your positive feedback.
>
>
>
> **Weaknesses**
>
>
> **Further clarification of Table 3**
>
> At test time, both parametric and non-parametric methods need resources for computation to compute gradients (parametric) or to adjust weights (non-parametric). We therefore believe that Table 3 is essential since it quantifies computational costs such as storage space (1.4 GB for all), number of trainable parameters, and TFlops. Our method performs simple matrix computations for which we will include the CPU cost consumed in the paper. Given MICCAI's focus this year on low-resource settings, we believe that specifying the exact requirements for our model to function effectively is necessary.
>
>
>
> **Calibration error**
>
> We report the ECE error by using the library developed in [I] and calculate the ECE error between the predicted labels and true labels, which is useful for uncertain environments. We mainly compared the ECE score for parametric methods and our method to quantify the calibration of models in uncertain scenarios, as demonstrated in [II]. However, we will also include the ECE error for T3A to the paper. Additionally, we will include ECE errors for additional backbones and methods in future work.
>
>
>
> **Feedback**
>
>
>
> **Including quantitative and additional benefits of our method in abstract**
>
> We will strengthen the abstract by including the results of classification accuracy, runtime, and ECE error. Thank you for pointing this out.
>
>
>
> **Neighborhood information**
>
> We use neighborhood information of the source prototypes with the target data at test-time. We will add this in the introduction.
>
>
>
> **Figure 1**
>
> We will modify the Figure to include a comprehensive overview of the neighborhood exploration with the distance functions as suggested; thank you for this suggestion.
>
>
>
> **Further explanation of prototypes**
>
> We will include further details on how the prototypes are calculated at test time to the paper, within the space limitations.
>
>
>
> References:
>
> [I] Webpage: [https://torchmetrics.readthedocs.io/en/v0.8.0/classification/calibration_error.html](https://torchmetrics.readthedocs.io/en/v0.8.0/classification/calibration_error.html)
>
> [II] Kumar, Ananya, Percy S. Liang, and Tengyu Ma. "Verified uncertainty calibration.", NeurIPS 2019

---

### Official Review · Reviewer_ugvA · 2024-07-08

**Recommendation:** 3
**Confidence:** 4

**Clarity:**

The paper is generally clear but has some clarity issues that could be addressed with moderate revision

**Feedback:**

Overall, the paper is interesting. However, it would benefit from a revision of the methodology to ensure the proposed approach is sufficiently described, including an illustration of the methodology. Please also see the weakness section for more feedback.

**Justification:**

Multiple weaknesses were identified, such as an insufficient presentation of the methodology, arbitrary choices of competitors, and a lack of significance tests. By addressing these points, the submission could be substantially improved, and the paper could possibly be accepted, but not in its current state.

**Reproducibility:**

Sufficient amount of details available for reproducing the main results, but open access is not provided to source code and/or data

**Strengths:**

1. The proposed non-parametric method is novel and effectively addresses the computational inefficiencies associated with traditional fine-tuning approaches.
2. The availability of anonymized code makes the evaluation transparent.

**Summary:**

The paper presents an approach to test-time generalization using a non-parametric method named Test-time Non-parametric Neighbors (TNN) as opposed to fine-tuning, thus offering computational efficiency.

**Weaknesses:**

1. The paper lacks sufficient detail in the description of the TNN method. Key aspects such as the algorithmic steps, mathematical formulations, and implementation specifics are not comprehensively covered. The dynamic voting mechanism for selecting neighbors needs more clarity. The criteria for neighbor selection and the weight adjustment process should be elaborated. Overall, this makes it difficult for readers to fully understand. This weakness, for example, could be solved by illustrating the method in sufficient detail, which could replace Fig. 1, as its information content is very limited.
2. Can the authors explain the direct relation of their method to the Johnson-Lindenstrauss lemma? What kind of assumptions are needed for the lemma to hold? Are there any constraints that need to be incorporated during training in order to ensure a stable local neighborhood?
3. The selection of competitors for evaluation, especially for Tables 2 and 3, appears arbitrary. For example, the competitor in Table 2 (Tent) seems to achieve the lowest performance for the respective backbone. It is crucial to justify why specific methods were chosen and ensure a comprehensive comparison with the most competitive approaches when only considering single techniques, as in Table 2.
4. While using Messidor and Fetal-8 datasets is appropriate, the paper should include a wider variety of datasets to better demonstrate the proposed method's generalizability. For example, the paper lacks an evaluation of large-scale datasets and a discussion on scalability.
5. The results presented lack statistical analysis to support the claims. Including standard deviation, confidence intervals, or p-values would strengthen the evaluation.

---

> ### Author Response · Authors · 2024-07-12
> **Rebuttal by Authors**
>
> We would like to thank the reviewer for their constructive suggestions, which we believe have contributed to an improved manuscript.
>
>
>
> **1. Additional details of TNN**
>
> We obtain neighbors utilizing the cosine distance function and further select appropriate neighbors via dynamic voting. For dynamic voting, we calculate the mean of obtained classifier weights through different permutation sets of top neighbors that are chosen by their closeness, measured by the distance function to predict the classification labels. We will add this clarification to the paper. To enhance reproducibility, we have open-sourced the code. We agree with the reviewer that replacing Figure 1 with an illustration will add more insights and will be added to the paper. Moreover, a pseudo-code reflecting the algorithmic steps of our model will be added to the paper.
>
>
>
>
> **2. Further clarification about JL Lemma**
>
> The JL lemma provides a theoretical foundation for our work. It does not impose any restrictions on our method. Specifically, the lemma ensures that if points lie close to each other in higher dimensional space, then, even when projected randomly, these points will also lie close to each other in a lower dimensional space [I]. In our case, the projections are better than random due to models being trained on the source domains.
>
>
> Pandey et al., 2020 [II] propose an alternate lemma. In their work, they utilize gradient descent-based optimization to sample neighbors from the VAE latent space. Different from them and other methods, we obtain neighbors without any optimization, thus making the adaptation process at test-time entirely feedforward. We will include this discussion in the paper.
>
>
>
>
> **3. Baselines comparison**
>
> All baseline models chosen for comparison depict well-established SOTA approaches in terms of test time adaptation (TTA). Specifically, we have chosen baselines similar to T3A, as our model follows a similar direction, i.e., being non-parametric.
>
>
> Tent also depicts a well-established TTA method. As detailed in Section 3, Tent obtains the target model by estimation of the MAP solution. This depicts an error-prone approach, as laid out in our paper. Moreover, Tent's sole reliance on entropy minimization typically necessitates a larger batch size and more samples in the target dataset, which is not feasible in medical imaging scenarios, leading to suboptimal performance. Nevertheless, we have chosen to include Tent as a baseline model to demonstrate its insufficiency when applied to medical imaging.
>
>
>
>
> **4. Further clarification about extending to more datasets**
>
> We agree with the reviewer that the inclusion of more datasets will provide even more insights into our approach’s generalizability. However, we cannot add additional datasets due to the review/rebuttal guidelines of MICCAI and plan to include more datasets in future journal work.
>
>
>
>
>
> **5. Further clarification of our results**
>
> We thank the reviewer for pointing out the missing standard deviation values. As mentioned in the paper, we report the mean values by utilization of 5 random seeds. Hence, as suggested, we will also include the standard deviations for the baselines in the main table. However, this did not alter our findings and we will add this to the paper.
>
>
>
>
> **References:**
>
> [I] Ghojogh, Benyamin, Mark Crowley, Fakhri Karray, and Ali Ghodsi. Elements of dimensionality reduction and manifold learning. Springer, 2023.
>
> [II] Pandey, Prashant, et al. "Unsupervised domain adaptation for semantic segmentation of NIR images through generative latent search.", ECCV 2020

---

### Official Review · Reviewer_czyw · 2024-07-10

**Recommendation:** 5
**Confidence:** 3

**Clarity:**

The paper is clear and well-written, with minor areas for improvement in clarity

**Feedback:**

The paper is clearly written and the method is well explained. The challenge addressed by this work is particularly relevant in medical image analysis. I don't have any particular feedback, maybe a figure giving an overview of the proposed methodology could improve the clarity of the paper.

**Justification:**

Even though the overall numeric improvement is not that significant compared to other available techniques, the proposed method is interesting and versatile, as it can be applied to any trained classifier.

**Reproducibility:**

Sufficient amount of details available for reproducing the main results, but open access is not provided to source code and/or data

**Strengths:**

The authors propose a novel formulation of a non-parametrich time-test adaptation technique.
The method in the paper presents a robust approach to test-time generalization that does not require modifications to the source training procedure, making it versatile for use with any pre-trained model. It leverages empirical risk minimization (ERM) to enhance model generalization under distribution shifts. At test time, the method uses a non-parametric approach, computing source prototypes and utilizing nearest neighbors for classification.
This approach is efficient, scalable, and easy to implement, with flexibility in choosing distance metrics, making it a practical solution for real-time applications and various data domains.
The authors carry out careful and thourough testing, applying their method to three different backbones and test it against several parametric and non-parametric baseline. They validate the models on two publicly available datasets.
The results are reproducible, the authors have given access to anonymised code and use publicly available datasets.

**Summary:**

The authors propose a novel non-parametric test-time adaptation technique to improve generalisability of deep learning models affected by domain shifts. They apply their approach on three different backbones and validate it against several time test adaptation techniques from the state of the art on two publicly available datasets.

**Weaknesses:**

The method's weaknesses include its dependence on pre-trained models and the assumption that the source model separates classes well. Scalability issues may arise with high-dimensional data or large datasets, and handling noisy data remains a challenge despite dynamic voting. The initial prototypes might be suboptimal if the source and target domains differ significantly, and the method is sensitive to the choice of distance metrics. The lack of end-to-end optimization during inference can limit adaptability, while iterative updates may increase computational overhead. Additionally, the dynamic voting mechanism adds complexity, and the method's success is heavily influenced by the quality of the source domain data.
Moreover, the imrpovement obtained compared to the baseline methods is not always significantly higher.

---

> ### Author Response · Authors · 2024-07-12
> **Rebuttal by Authors**
>
> We thank the reviewer for their constructive suggestions, which we believe have contributed to an improved manuscript.
>
>
> **Source model training and class separability**
>
> The dependence on the utilization of a pre-trained model for source training is a common requirement of test time adaptation methods, including T3A, Tent, and SHOT. At test-time, our non-parametric approach explores neighborhood information and directly obtains the target model $\theta_{t}$ through classifier adjustments. This results in a target-specific model that utilizes target information more effectively than other methods and facilitates class separability.
>
>
>
> **Prototypes for larger domain shifts**
>
> For the computation of prototypes, we rely on source model predictions, such as T3A, and employ the distance function to find neighbors based on the JL Lemma. The JL Lemma asserts that points will remain close to each other even after random projections. In our case, our projections are better than random projections, which allows the distance function to identify neighbors. Significant domain shifts remain an open problem for all test-time adaptation methods.
>
>
>
> **Why end-to-end optimization is not useful**
>
> We avoid end-to-end optimization to obtain the target model, aside from increased computational usage, for two main reasons:
>
> i) MAP Estimation: It is not reliable for adaptation. (As detailed in Methodology)
>
> ii) Iterative Re-Training with gradient-based optimization: Since not all the predictions are accurate, retraining on these noisy predictions will lead to a misspecified model [1].
>
>
>
> **Dynamic voting and performance**
>
> Dynamic voting enables us to identify the most useful neighbors, which is crucial for the effective updating of prototypes across the streaming batches of data at test time. Without this, updating prototypes would be less beneficial. Across three different backbones, our method consistently achieves the best (in most scenarios) or second-best performance.
>
> We would like to thank the reviewer for suggesting to depict an overview figure. As also suggested by Reviewer rJeC, we will revise and improve Figure 1 accordingly.
>
>
>
> **References:**
>
> [1] Ba, Cuimin. "Robust Misspecified Models and Paradigm Shifts." arXiv 2021.

---

### Official Review · Reviewer_U4mh · 2024-07-11

**Recommendation:** 3
**Confidence:** 4

**Clarity:**

The paper is generally clear but has some clarity issues that could be addressed with moderate revision

**Feedback:**

1.	To demonstrate why this method is state-of-the-art for test-time generalization, it would benefit from experiments to demonstrate why certain decisions were made and why this is better than current methods. For example, ablation studies on why cosine similarity is used as the distance metric, or experiments which demonstrate the improvement of performance by not using all of the neighbors for finding pseudo-labels.
2.	The work would benefit from a more thorough literature review of current methods. This would not only provide a stronger motivation for your method but could also assist in developing a more innovative approach, or in articulating why your method is considered novel.
3.	The writing of the methods section is quite hard to follow. You don’t want to lose readers, so I would recommend making the writing clearer so that your message is more easily conveyed.

**Justification:**

To change my judgement, I would need more convincing on the novelty of the work. The paper is also missing ablation studies to explain why decisions were made when designing this approach, without this it is hard to understand why this method would be beneficial.

**Reproducibility:**

Sufficient amount of details available for reproducing the main results, and open access is provided (or promised upon acceptance) to source code and/or data

**Strengths:**

1.	The authors have identified an interesting problem to research. Application of deep learning models in clinical settings requires the models to generalize to domain-shifted data. Non-parametric methods for test-time generalization show potential because they are less computationally expensive than other approaches, making them useful for real-time medical applications.
2.	The paper tests three different backbones for two different medical settings, which is good for demonstrating the utility of their proposed method.
3.	The paper has good figures which clearly explain the difference between parametric and non-parametric techniques.
4.	The paper utilizes public datasets and has released its code, facilitating the reproduction and validation of results.

**Summary:**

Recognising that test-time generalization techniques which involve fine-tuning gradients or batch-norm statistics are computationally expensive (and hence less useful for real-time medical applications), the paper suggests a non-parametric method for test-time generalization that doesn’t require fine-tuning or optimization. The paper suggests a prototype-based method, which calculates prototype representations to represent each class in the embedding space, and then predict the label of an input as the class of the nearest prototype. The method continually updates the class prototypes with new samples to account for distribution shifts. The method is tested on two medical image datasets, retinal images and fetal ultrasound images, and show better test-time generalization accuracy than other parametric and non-parametric methods.

**Weaknesses:**

1.	The novelty of this proposed method is questionable. Previous works, such as Jang, M. ‘Test-Time Adaptation Via Self-Training With Nearest Neighbor Information’ (2023), use a similar Prototype-based classification method. This includes using cosine similarity in the embedding space to class prototypes for classification and filtering out unreliable examples when calculating nearest neighbor-based pseudo labels. The paper mentioned also demonstrates the benefit of nearest-neighbor information in an optimization-free case in their ablation study (see TASH-N). The paper requires more explanation as to why this method is novel, other than being applied to new datasets.
2.	The work would benefit from `ablation studies to help explain the choices made when designing the method and to support the claim it is better than current state-of-the art methods. For example, the paper says, ‘we utilized the cosine distance here, while in principle every other distance metric can be use’. An ablation study would be useful to explain why using cosine similarity is better than using a different distance metrics (e.g. Euclidean distance, Mahalanobis distance etc) for this method.
3.	The methods section can be difficult to follow and understand. For example, the paper doesn’t really define what it means when it says it uses ‘dynamic voting’ and it doesn’t define some of the symbols used in equations (see section on JL Lemma).  A better approach might be to include more equations, as mathematical explanations can sometimes clarify the procedures more effectively.

---

> ### Author Response · Authors · 2024-07-12
> **Rebuttal by Authors**
>
> We thank the reviewer for their constructive suggestions, which we believe have contributed to an improved manuscript.
>
> **Novelty and comparison**
>
> As stated in our introduction and also mentioned by the reviewer, TAST [10] uses neighborhood information to perform test-time adaptation. However, they utilize backpropagation for adaptation, a procedure that we are aiming to avoid due to the reasons discussed in Section 3 of our paper. Furthermore, multiple MLP ensemble modules are employed by TAST, which is not required by our model. Additionally, no theoretical foundation for their method is provided. TAST-N (from the ablation study of [10]) uses feedforward adaptation to obtain classifier weights via the usage of cosine similarity. However, our method uses subsets of neighbors and employs dynamic voting and obtain the final weights of the classifier. This ensures more robustness and a better-aligned target-specific model compared to TAST-N, which computes the weights directly. We will include this discussion in the paper.
>
>
>
> **Ablation studies**
>
> Thank you for suggesting new experiments to ablate the distance functions. However, we cannot add additional results due to MICCAI's review/rebuttal guidelines, and due to space limitations of the workshop format, we concentrated on the main comparisons to baseline models. We will certainly include insights from ablations in our future journal work.
>
>
>
> **Additional details of our method**
>
> For dynamic voting, we calculate the mean of obtained classifier weights through different sets of top neighbors that are chosen by their closeness, measured by the distance function, to predict the classification labels. We will include this discussion and pseudocode in the paper.
>
>
>
> **Additional literature review**
>
> Thank you for your feedback; while no specific papers were cited, we will expand our literature review with additional recent studies.

---

### Meta-Review · Area_Chair_MMBD · 2024-07-16

**Recommendation:** Accept (Oral)
**Confidence:** 4

**Metareview:**

The paper proposes a novel and potentially useful non-parametric test-time adaptation (TTA) method, TNN, for medical image analysis. The method achieves good performance on multiple datasets and outperforms current state-of-the-art approaches in terms of accuracy, runtime, and calibration error. The reviewers find the approach efficient, scalable, and easy to implement. They also acknowledge thorough evaluation on multiple datasets and architectures. Consider adding the ablation studies in future work to further strengthen the work. Also, evaluating the method on a wider variety of datasets, especially large-scale ones, would be beneficial. The authors have promised to improve the clarity of the TNN method description with pseudocode and visualizations and to add the standard deviation in the results for all models. Abstract and introduction needs to be strengthened with a clearer explanation in the final version. As discussed in the rebuttal, sufficient technical details must also be added to the final paper.

---

### Decision · Program_Chairs · 2024-07-16

Accept (Oral)